# A Reparametrization-Invariant Sharpness Measure Based on Information Geometry

**Cheongjae Jang**
Hanyang University
cjjang@hanyang.ac.kr

**Sungyoon Lee**
Korea Institute for Advanced Study
sungyoonlee@kias.re.kr

**Frank C. Park**
Seoul National University
Saige Research
fcp@snu.ac.kr

**Yung-Kyun Noh**
Hanyang University
Korea Institute for Advanced Study
nohyung@hanyang.ac.kr

## Abstract

It has been observed that the generalization performance of neural networks correlates with the sharpness of their loss landscape. Dinh et al. (2017) [8] have observed that existing formulations of sharpness measures fail to be invariant with respect to scaling and reparametrization. While some scale-invariant measures have recently been proposed, reparametrization-invariant measures are still lacking. Moreover, they often do not provide any theoretical insights into generalization performance nor lead to practical use to improve the performance. Based on an information geometric analysis of the neural network parameter space, in this paper we propose a reparametrization-invariant sharpness measure that captures the change in loss with respect to changes in the probability distribution modeled by neural networks, rather than with respect to changes in the parameter values. We reveal some theoretical connections of our measure to generalization performance. In particular, experiments confirm that using our measure as a regularizer in neural network training significantly improves performance.

## 1 Introduction

From recent discoveries in deep learning, it has been conjectured that the generalization performance of neural networks correlates with the sharpness of their loss landscape. In particular, lower generalization performance of large-batch training of stochastic gradient descent (SGD) has been attributed to its convergence to sharper minima [22, 49], and several optimization methods have shown better generalization performance by actively finding solutions with lower sharpness [7, 19, 4, 43, 12, 24]. Minimum description length (MDL)-based arguments [16, 17] and the generalization upper bound of the PAC-Bayes theory for neural networks [10] also suggest that the flatter the loss landscape, the better the generalization performance. Furthermore, in [21, 32], measures based on the sharpness concept show better performance in predicting the generalization performance of models among various generalization measures considered in the past including the margin [5] and norm [34, 33].

Various measures for sharpness and flatness have been proposed in the literature. Hochreiter & Schmidhuber (1997) [17] propose the concept of 'flat minima,' in which the flatness of a minimum is interpreted as the volume of the connected region in the parameter space over which the loss has approximately similar values. Similarly, the sharpness has often been measured by the maximum loss value (inside a Euclidean ball) near the minimum [22] or by the spectral norm of the Hessian at the minimum [49].

36th Conference on Neural Information Processing Systems (NeurIPS 2022).

The above definitions of sharpness have some notable critical flaws. Dinh et al. (2017) [8] point out that certain parameter scalings (e.g., for neural networks with ReLU activation functions [31]) and reparametrizations may have no effect on the model output or overall generalization performance, and yet lead to wildly different values of sharpness.[1] Other sharpness measures have been proposed to address the lack of scale-invariance [26, 46, 40, 39], but these measures fail to be invariant with respect to reparametrizations; Section 5 of [8] offers several nonlinear reparametrization examples. The connections between sharpness and generalization performance also remain elusive.

In this paper we address the lack of scale- and reparametrization-invariance of existing sharpness measures, as well as the problem that they often do not explain the generalization performance of deep learning networks nor lead to practical use to improve the performance. Our approach rests on the observation and insight that sharpness should be measured taking into proper account the geometry of the underlying parameter space. For this purpose we draw upon tools from information geometry, paying attention to the fact that in most classification problems, probabilistic classification models are parametrized in terms of neural networks. In information geometry, the Fisher information matrix (FIM) serves as a natural Riemannian metric for the parameter space [3] and allows for measuring the change in probability density with respect to changes in the parameter values.

The eigenspectra of the FIM of a neural network are observed to have a small number of positive outliers (the number is usually equated with the number of classes) and a bulk consisting of small eigenvalues [41, 37, 36, 48, 14]. This observation indicates that the number of principal varying components of the probability densities modeled by neural networks is significantly lower compared to the number of parameters. Therefore, identically weighting every possible parameter change direction in the parameter space – that is, applying Euclidean geometry – can be problematic, since it places too much weight on parameter change directions that are meaningless with respect to changes in the probabilistic model, or equivalently, not enough weight is placed along meaningful directions.

We argue that one should consider these implications and use the change in probabilistic models as a 'ruler' when defining a sharpness measure of neural network loss landscapes that does not depend on how the model is parametrized.

Building on the above geometric analysis of the neural network parameter space, in this paper we propose a new sharpness measure based on information geometry. This measure is by definition reparametrization-invariant. Additionally, we show that this sharpness measure simultaneously satisfies the scale-invariance for neural networks with ReLU activation functions, since it is invariant for parameter transforms that do not change the model output. Hence our measure is free from all the reparametrization- and scale-variance issues of previous sharpness measures raised by [8].

As a second contribution of this paper, we also show that our geometric sharpness measure sheds an important insight on generalization performance. As detailed in Section 3, when the FIM is replaced with the Hessian in our measure, the measure is in a form similar to Takeuchi's information criterion (TIC), which quantifies the asymptotic bias of the log-likelihood value (usually the negative loss) evaluated at the maximum likelihood estimates hence corresponding to the expectation of the generalization gap [44]. TIC for neural networks has empirically shown a strong correlation with the generalization gap [45]. One can apply the same argument to our measure since the Hessian can be approximated by the FIM under mild conditions for deep neural networks [28]. In addition, we show that our measure can be linked to generalization in another geometrical way. The measure is associated with a margin defined in an information geometric sense, enabling the interpretation that a smaller sharpness measure indicates larger margins. We also demonstrate experimentally that using our measure as a regularizer to train neural networks can significantly improve generalization performance.

Our contributions can be summarized as follows:

- We provide an information geometric analysis of the neural network parameter space by investigating the eigensubspace of the Fisher information matrix (FIM) of neural networks.
- We propose a reparametrization- and scale-invariant sharpness measure based on information geometry that is free from the problems of existing sharpness measures posed by [8], and discuss the relation between the measure and generalization performance.

---

[1]In this paper, unlike other works where parameter scalings are often referred to as (linear) reparametrizations, following [8], it is called a reparametrization to represent a model on a different parameter space obtained by applying a (nonlinear) bijection to the original parameters.

- We use the proposed measure as a regularizer when training neural networks, and demonstrate improved generalization performance.

We provide an information geometric analysis of the neural network parameter space in Section 2. Building on this analysis, Section 3 presents our reparametrization-invariant sharpness measure based on information geometry and connects the proposed measure to generalization. In Section 4, we perform numerical experiments using our measure as a regularizer to improve generalization performance in neural network training.

## 2  An information geometric analysis of the neural network parameter space

In this paper, we focus on the probabilistic classification models parametrized by neural networks. Let $x \in \mathbb{R}^D$ denote data, and $y \in \{1, \dots, C\}$ denote class labels with $C$ as the number of classes. Let $p(x, y)$ and $p(y|x; \theta)$ respectively denote the data generating distribution and the parametric model of the probability density of classes given data. Here the parameter $\theta \in \mathbb{R}^m$ is the set of all the weight and bias parameters of neural networks. We consider a neural network as a function $f(\cdot; \theta) : \mathbb{R}^D \to \mathbb{R}^C, x \mapsto f(x; \theta) = (f_1(x; \theta), \dots, f_C(x; \theta))^\top$, where $f_j(\cdot; \theta) : \mathbb{R}^D \to \mathbb{R}$ is a function that returns the $j$-th logit for $j = 1, \dots, C$. Our parametric model is then represented as $p(y|x; \theta) = \frac{\exp(f_y(x))}{\sum_j \exp(f_j(x))}$, and we denote the negative log-likelihood by $l(y, f(x; \theta)) = -\log p(y|x; \theta)$.

Suppose data $\{(x_1, y_1), \dots, (x_N, y_N)\}$ are drawn i.i.d. from the data generating distribution $p(x, y)$. The cross-entropy loss, our training objective function throughout the paper, can be written as follows:[2]

$$\mathcal{L}(\theta) = \frac{1}{N} \sum_{i=1}^{N} l_i(\theta), \tag{1}$$

where $l_i(\theta) = l(y_i, f(x_i; \theta)) = -\log p(y_i | x_i; \theta)$.

### 2.1  The Fisher information matrix (FIM)

The Fisher information matrix (FIM) for the family of probability density functions $p(x, y; \theta) = p(x)p(y|x; \theta)$ parametrized by the neural network parameters $\theta$ is defined as

$$F(\theta) = \mathbb{E}_{p(x)} \left[ \mathbb{E}_{p(y|x;\theta)} \left[ \left(\frac{\partial l}{\partial \theta}\right)^\top \left(\frac{\partial l}{\partial \theta}\right) \right] \right] = \mathbb{E}_{p(x)} \left[ J(x)^\top \mathbb{E}_{p(y|x;\theta)} \left[ \left(\frac{\partial l}{\partial f}\right)^\top \left(\frac{\partial l}{\partial f}\right) \right] J(x) \right]$$

$$= \frac{1}{N} \sum_{i=1}^{N} J_i^\top \left( \mathrm{diag}(p_i) - p_i p_i^\top \right) J_i, \tag{2}$$

where the dependence of $l(y, f(x; \theta))$ to $y, f(x; \theta)$ is omitted for simplicity and $J(x) = \frac{\partial f}{\partial \theta}(x) \in \mathbb{R}^{C \times m}$ is the Jacobian of the logits with respect to the parameters. In deriving (2), the expectation with respect to $p(x)$ is approximated by the finite sum over data points $x_1, \dots, x_N$ drawn i.i.d. from $p(x)$, $J_i = J(x_i)$, and we use the fact that $\mathbb{E}_{p(y|x_i;\theta)} \left[ \left(\frac{\partial l}{\partial f}\right)^\top \left(\frac{\partial l}{\partial f}\right) \right] = \mathrm{diag}(p_i) - p_i p_i^\top$, where $p_i = (p(1|x_i; \theta), \dots, p(C|x_i; \theta))^\top \in \mathbb{R}^C$ is the model's prediction on the probability that $x_i$ belongs to each class and $\mathrm{diag}(p_i) \in \mathbb{R}^{C \times C}$ is a diagonal matrix whose $(j, j)$ entry is $(p_i)_j$.

In this paper, among many of the important properties and applications of the FIM (e.g., being the Cramer-Rao lower bound of estimator variances in estimation theory and statistics), we focus on the fact that the FIM can serve as a metric to measure the distance between the parametric probability density models in information geometry. Differential geometrically speaking, the FIM acts as a natural Riemannian metric on the statistical manifold, a space where the family of probability densities modeled with smoothly varying parameters $\theta \in \mathbb{R}^m$ is gathered.[3] The FIM makes it possible to

---

[2]We discuss the applicability of our measure to the square loss in Appendix F.

[3]Note that the family of probabilistic models from neural networks is usually not a manifold in a mathematically rigorous sense due to the inherent singularities in the neural network parameter space (see Section 12.2 of [3]), but this fact is not crucial for our discussion.

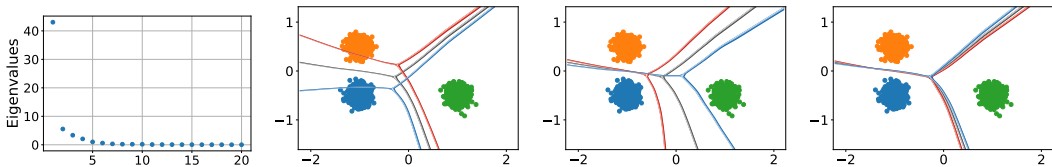

(a) Top 20 eigenvalues of $F(\theta)$

(b) Perturbed decision boundaries from $\Delta\theta_1$

(c) Perturbed decision boundaries from $\Delta\theta_2$

(d) Perturbed decision boundaries from $\Delta\theta_{10}$

Figure 1: A synthetic three-class classification example. For (b)-(d), the black, red, and blue lines correspond to decision boundaries of the neural network with the trained parameter values, and parameter values perturbed along the $k$-th eigenvector $\Delta\theta_k \in \mathbb{R}^m$ (associated with the $k$-th eigenvalue) of $F(\theta)$ with steps of 0.5 and -0.5, respectively.

measure geometric quantities such as length, angle, and volume on the manifold. (We refer the reader to [6, 11] for the backgrounds on the Riemannian manifolds and differential geometry, and [3, 35, 28] for those on the information geometry and the Fisher information matrix.) Note that when the FIM is not full-rank, e.g., in the case of overparameterized neural networks with $m \gg N$, it can be used as a pseudo metric.

The FIM is closely related to the KL-divergence, which is frequently used as an information theoretic measure of discrepancy between probability density functions. The KL-divergence between two probability density functions with infinitesimal parameter difference $d\theta \in \mathbb{R}^m$ can be approximated as

$$\mathrm{KL}(p(x,y;\theta+d\theta)||p(x,y;\theta)) \approx \frac{1}{2}d\theta^\top F(\theta)d\theta. \tag{3}$$

## 2.2   An analysis of the eigensubspace of the FIM

This section discusses the characteristics of the eigensubspace of FIM of neural networks. In the analysis of the eigenspectra of deep neural network Hessians of [37], the FIM in (2) has been decomposed as follows:

$$F(\theta) = \frac{1}{N}\sum_{i=1}^{N} J_i^\top \left(\mathrm{diag}(p_i) - p_i p_i^\top\right) J_i = \frac{1}{N}\sum_{i=1}^{N} J_i^\top M_i^\top M_i J_i, \tag{4}$$

where $M_i = \mathrm{diag}(\sqrt{p_i})\left(I - \mathbb{1}p_i^\top\right) \in \mathbb{R}^{C \times C}$, $\sqrt{p_i} = (\sqrt{(p_i)_1}, \dots, \sqrt{(p_i)_C})^\top \in \mathbb{R}^C$, and $\mathbb{1} = (1, \dots, 1)^\top \in \mathbb{R}^C$ is a vector whose elements are all one.

According to the decomposition, the FIM can be thought of as a non-centralized second moment of the (modified) logit gradients, i.e., each row of $M_i J_i \in \mathbb{R}^{C \times m}$ whose $j$-th row vector is represented as

$$(M_i J_i)_{j:}^\top = \sqrt{(p_i)_j} \cdot \left((J_{i,j})^\top - \sum_{j'=1}^{C}(p_i)_{j'}(J_{i,j'})^\top\right) \in \mathbb{R}^m, \tag{5}$$

where $J_{i,j} \in \mathbb{R}^{1 \times m}$ is the $j$-th row of $J_i$. It has been observed that these logit gradients form a kind of hierarchical structure after sufficient training. In the observation from [36], the gradients of the $c' \neq c$-th logit calculated from the data in class $c$ are gathered to form a cluster. A matrix obtained from averaging the outer products of the averaged logit gradients belonging to each cluster is then attributed to the outliers of the eigenspectra, and the outliers of the eigenvalues appear as much as the number of classes. That is, the principal eigenvalues of the FIM (and possibly the eigenvectors) can be closely connected with distinguishing each class $c$ from the rest of the $c' \neq c$ classes (as can be implied from averaging (5) for $j \neq c$ and by assuming $(p_i)_c \approx 1$). This becomes more evident in Appendix A, which assumes the prediction is balanced as $(p_i)_{c'} = \epsilon$ for all $c' \neq c$ with $\epsilon \ll 1$.

In order to more intuitively understand the characteristics of the eigensubspace of FIM implied from the above analysis, we provide a classification example for two-dimensional synthetic data generated from mixtures of three Gaussians in Figure 1. We trained a three-layer fully connected neural network with 70-dimensional hidden units (the number of parameters = 5,393).

Figure 1 (a) shows the top twenty eigenvalues of the FIM of the trained neural network. Note that there are few outliers among the eigenvalues in the entire 5,393-dimensional parameter space, with the remainder being almost 0. The eigenvalues indicate the rate of change of the probability density (modeled by the neural network) measured by the KL-divergence for a unit change of parameter values along the corresponding eigenvectors. The existence of a few outliers means that the density change mainly occurs in only a few specific directions and that the amount of change in probability density can vary significantly according to the direction of change in parameter values.

This fact can also be observed in decision boundaries obtained from neural networks with the parameter values perturbed along some eigenvectors shown in Figure 1 (b-d). When perturbing the values along the principal eigenvectors, a significant change appears in the decision boundary, increasing or decreasing the margin of certain classes. On the other hand, for eigenvectors with small eigenvalues, there is almost no change in the decision boundary according to the same level of change in parameter values (in the Euclidean sense).

These examples imply that it would be more meaningful to reflect the different influences on the probabilistic model according to the change direction when we measure the quantities concerning changes in the parameter values, e.g., the sharpness of the loss landscape. By reflecting on these implications and using the unit change in probability distribution modeled by neural networks as our 'ruler' to measure the sharpness, that is, applying the information geometry, the following section presents a sharpness measure that does not rely on how the model is parametrized.

## 3 An information geometric sharpness measure

Reflecting the above analysis, we define an information geometric sharpness (IGS) measure as follows:

$$\text{IGS}(\theta) = \frac{1}{N} \sum_{i=1}^{N} \left( \frac{\partial l_i}{\partial \theta} \right) F(\theta)^\dagger \left( \frac{\partial l_i}{\partial \theta} \right)^\top, \tag{6}$$

where $F(\theta)^\dagger \in \mathbb{R}^{m \times m}$ is the pseudo-inverse of $F(\theta)$ and $l_i = l(y_i, f(x_i; \theta))$ is the cross-entropy loss evaluated at the $i$-th data $(x_i, y_i)$.

Our measure evaluates the squared norm of the loss gradients calculated at each data point using the (pseudo-inverse of) FIM and averages them for all data points.[4] Since $\left( \frac{\partial l_i}{\partial \theta} \right)^\top \in \mathbb{R}^m$ belongs to the span of the eigenvectors of $F(\theta)$ with non-zero eigenvalues, our sharpness measure is well-defined in the sense that there are not any components of $\left( \frac{\partial l_i}{\partial \theta} \right)$ neglected in measuring its information geometric norm (we elaborate on this in Appendix B).

Our sharpness measure in (6) is an intrinsic quantity, i.e., coordinate-invariant (in differential geometric terms). To see why, observe that under a local coordinate transformation (or a reparametrization) $\phi : \mathbb{R}^m \to \mathbb{R}^m$, $\theta \mapsto \theta' = \phi(\theta)$, i.e., the model is now represented with respect to a different parameter $\theta'$ as $\tilde{f}(\cdot; \theta') = f(\cdot; \phi^{-1}(\theta')) = f(\cdot; \theta)$, $F$ and $\left( \frac{\partial l_i}{\partial \theta} \right)$ transform according to the following rules: (i) $F \mapsto F' = \Phi^{-\top} F \Phi^{-1}$, where $\Phi = \frac{\partial \phi}{\partial \theta} \in \mathbb{R}^{m \times m}$; (ii) $\left( \frac{\partial l_i}{\partial \theta} \right) \mapsto \left( \frac{\partial l_i}{\partial \theta'} \right) = \left( \frac{\partial l_i}{\partial \theta} \right) \Phi^{-1}$, where it can be verified that the $\left( \frac{\partial l_i}{\partial \theta} \right) F(\theta)^\dagger \left( \frac{\partial l_i}{\partial \theta} \right)^\top$ remains the same. Since the above measure gives the same value regardless of which parametrization (e.g., $\theta$ or $\theta'$) the statistical model is parametrized, it is free from the reparametrization-variance issues raised in [8].

We can also define other reparametrization-invariant measures by measuring the information geometric norm of the gradients of the entire loss or the mini-batch losses. For mini-batches of size $b$, the corresponding measure can be defined as follows:

$$\text{IGS}_b(\theta) = \mathbb{E}_{p(B)} \left[ \left( \frac{1}{b} \sum_{x_i \in B} \frac{\partial l_i}{\partial \theta} \right) F(\theta)^\dagger \left( \frac{1}{b} \sum_{x_i \in B} \frac{\partial l_i}{\partial \theta} \right)^\top \right], \tag{7}$$

where $\mathbb{E}_{p(B)}[\cdot]$ denotes the expectation with respect to some distribution $p(B)$ of mini-batches $B \subset \{x_1, \ldots, x_N\}$ of size $|B| = b$. We discuss the relation between (6) and (7) in Appendix C. Note

---

[4]The reason for defining the sharpness measure as in (6) will be evident in Section 3.3 where we develop the relationship between our measure and the generalization.

that this definition can be linked to the concept of m-sharpness, which measures the sharpness of the loss landscape by averaging the sharpness of mini-batch losses and shows a better correlation to the generalization as the batch size reduces [12].

## 3.1 Transformation invariance

Neural networks composed of activation functions such as ReLU and leaky ReLU possess scale-invariant properties. In the case of layer-wise scaling, for consecutive layers of linear transforms and ReLU, e.g., $f(x; \{W_1, \ldots, W_L\}) = W_L \cdot \text{ReLU}(W_{L-1} \cdots \text{ReLU}(W_1 x))$, where $W_l \in \mathbb{R}^{d_l \times d_{l-1}}$ for $l = 1, \ldots, L$ with $d_0 = D$ and $d_L = C$, if each weight $W_l$ is multiplied by a constant $c_l > 0$ and the constants satisfy $\prod_{l=1}^{L} c_l = 1$, there is no change in the outputs of the neural network under the same input values. A similar concept includes the node-wise scaling, multiplying weights entering a node (or a hidden variable in neural networks) by a positive constant and dividing weights out of the node by the same constant.

These kinds of weight scaling can be viewed as a transformation (or a mapping between identical parameter spaces) that ensures that the neural network model maintains the same probability density function, i.e., remains equivalent. If a transformation is differentiable and locally invertible without changing the probability density function that the neural network parameter models for all parameters on a given neighborhood $U \subseteq \mathbb{R}^m$ of $\theta$, our measures defined in (6) and (7) become invariant with respect to such a transformation. This fact can be expressed as the following proposition:

**Proposition 3.1.** *Suppose there exist open subsets $U, V \subseteq \mathbb{R}^m$ and an invertible and locally differentiable transformation $g : U \to V$ that satisfies $f(x; \theta) = f(x; g(\theta))$ for all $x \in \mathbb{R}^D$ and $\theta \in U$ with $g(U) \subseteq V$. The information geometric sharpness measures in (6) and (7) are invariant to such a transformation, i.e., $\text{IGS}(\theta) = \text{IGS}(g(\theta))$ (or $\text{IGS}_b(\theta) = \text{IGS}_b(g(\theta))$) for all $\theta \in U$.*

The proof of Proposition 3.1 is provided in Appendix D.1.

Since the layer-wise and node-wise scalings considered for neural networks with ReLU satisfy the assumptions of Proposition 3.1, our measures in (6) and (7) are invariant to such scalings. Proposition 3.1 is more general than the scale-invariance, and Appendix D.2 provides examples of transformations other than parameter scalings that satisfy the assumptions in the proposition.

## 3.2 A comparison to some previous sharpness measures

After [8] pointed out the critical scale- and reparametrization-variance issues in the sharpness measures considered in [17, 22, 49], various scale-invariant sharpness measures have been proposed that suit to neural networks with activation functions satisfying the non-negative homogeneity conditions (i.e., $\sigma(a \cdot x) = a \cdot \sigma(x)$ for $a > 0$) such as ReLU [40, 39, 46, 26]. The measure presented in [40] is based on the geometry of the quotient manifold obtained from an equivalence class of neural networks, establishing the scale-invariance depending on the considered equivalence types. The work in [39] considers a layer-wise flatness measure for neural networks and additionally explores the conditions for the measure to explain the generaliza-

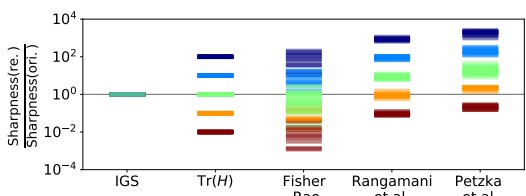

Figure 2: The ratio of sharpness measures evaluated for the reparametrized models to those for the original models ($\frac{\text{Sharpness (re.)}}{\text{Sharpness (ori.)}}$). Note that the y-axis is in the log scale. For the nonlinear reparametrization, we use $\eta = g(\theta) = (|\theta - \hat{\theta}|^2 + b)^a (\theta - \hat{\theta}) + \hat{\eta}$ considered in [8] with several choices of $(a, b)$, which are represented by different colors.

tion well. Also, a scale-invariant sharpness measure based on the PAC-Bayes theory is developed in [46]. The Fisher-Rao norm is proposed as a capacity measure for neural networks in [26] and defined as $\theta^\top F(\theta)\theta$, which is scale-invariant.

However, the above measures do not satisfy the invariance to general nonlinear reparametrizations, hence solving the problem raised by [8] only partially. That is, for a nonlinear reparametrization $\theta \mapsto \eta = g(\theta)$, the measures evaluated on the reparametrized model $\tilde{f}(\cdot; \eta) = f(\cdot; g^{-1}(\eta)) = f(\cdot; \theta)$ (with respect to $\eta$) can arbitrarily vary from those evaluated on the original model $f(\cdot; \theta)$ (with respect to $\theta$). In Figure 2, we empirically demonstrate the existence of this problem in conventional sharpness

measures such as the trace of the Hessian, the measures of [26], [40], and [39] (respectively denoted as $\mathrm{Tr}(H)$, Fisher-Rao, Rangamani et al., and Petzka et al. on the x-axis of the figure; see Appendix G.1 for details). We can observe that the magnitudes of these measures can vary significantly.

Compared with the previous measures, only our measure (denoted as IGS on the x-axis of Figure 2) is invariant to the considered reparametrizations. These experiments empirically demonstrate that our measure satisfies the reparametrization-invariance, hence providing a solution that properly resolves the issues posed by [8].

## 3.3 Connections to the generalization

### 3.3.1 Connections to Takeuchi's information criterion (TIC)

Our measures can be linked to generalization in different aspects. When we train parametric models via the maximum likelihood estimation (this is equivalent to minimizing the negative log-likelihood of (1)), the obtained maximum likelihood estimate is asymptotically unbiased, guaranteed theoretically by the asymptotic normality. However, the log-likelihood value (or the negative loss) evaluated at the obtained estimate does not enjoy such a property.

The well-known Akaike's information criterion (AIC) in the model selection literature calculates this bias under the assumption that the parametric model contains the actual data distribution [1]. Takeuchi's information criterion (TIC) is a generalized form of the AIC by considering the case of misspecified models, i.e., the parametric models do not contain the actual data distribution [44, 9, 45].

The bias $b(\hat{\theta})$ of the log-likelihood value evaluated at the maximum likelihood estimate is defined as follows:

$$b(\hat{\theta}) = \mathbb{E}_{\mathcal{D}} \left[ \frac{1}{|\mathcal{D}|} \sum_{(x_i, y_i) \in \mathcal{D}} \log p(x_i, y_i; \hat{\theta}(\mathcal{D})) - \mathbb{E}_{p(x,y)} \left[ \log p(x, y; \hat{\theta}(\mathcal{D})) \right] \right], \tag{8}$$

where $\mathcal{D} = \{(x_1, y_1), \ldots, (x_N, y_N)\}$ is the set of $N$ training data with $(x_i, y_i) \sim p(x, y)$ and $\hat{\theta}(\mathcal{D})$ denotes the maximum likelihood estimate obtained from using $\mathcal{D}$ [9]. Since our loss is the negative log-likelihood, the first and second terms inside $\mathbb{E}_{\mathcal{D}}[\cdot]$ correspond to the negative training and test losses, respectively. Consequently, the bias in (8) becomes the expectation of the parametric model's generalization gap (i.e., test loss – training loss).

TIC calculates this bias under asymptotic assumptions, i.e., in the limit $N \to \infty$, as follows [9]:

$$\mathrm{TIC} = \lim_{N \to \infty} b(\hat{\theta}) = \frac{1}{N} \mathrm{Tr}(H(\theta_0)^{-1} C(\theta_0)), \tag{9}$$

where $\theta_0 \in \mathbb{R}^m$ is a local maximum of the expected log-likelihood, $H(\theta_0) \in \mathbb{R}^{m \times m}$ is the Hessian of the (expected) loss, and $C(\theta_0) = \mathbb{E}_{p(x,y)} \left[ \left( \frac{\partial l(y, f(x;\theta))}{\partial \theta} \right)^{\top} \left( \frac{\partial l(y, f(x;\theta))}{\partial \theta} \right) \right]\Big|_{\theta=\theta_0} \in \mathbb{R}^{m \times m}$ is the non-centered covariance of the loss gradients. Here both $H(\theta_0)$ and $C(\theta_0)$ are evaluated using the data generating distribution $p(x, y)$.

The formulation in (9) is very similar to our information geometric sharpness measure when evaluated at a local minimum $\hat{\theta} \in \mathbb{R}^m$ of the loss in (1). Compared to TIC, our measure in (6) contains the FIM instead of the Hessian, and the expectation for $C(\theta)$ is taken with respect to the empirical distribution rather than the true distribution. After a model is sufficiently trained, the Hessian can be approximated to the FIM [36, 28], indicating that our sharpness measure is closely related to TIC.

It has been observed that TIC predicts the generalization gap of neural network models well when the expectation with respect to $p(x, y)$ is approximated by a finite sum of the integrands over the test data [45]. These findings are also confirmed by experiments using our measures. In Figure 3, compared to the other sharpness measures (labeled the same as in Figure 2), we can observe that our measure correlates better with the generalization gap. The experimental details are provided in Appendix G.2.

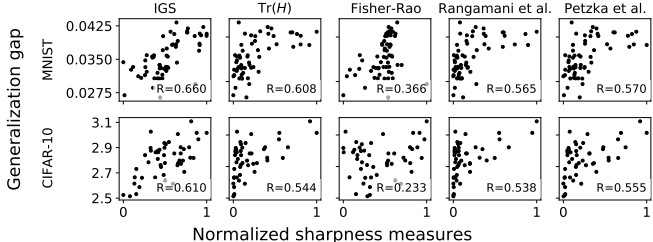
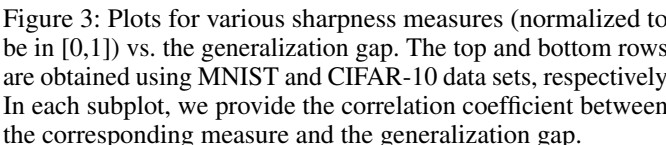

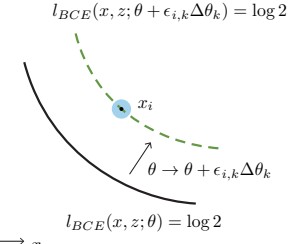

Figure 3: Plots for various sharpness measures (normalized to be in [0,1]) vs. the generalization gap. The top and bottom rows are obtained using MNIST and CIFAR-10 data sets, respectively. In each subplot, we provide the correlation coefficient between the corresponding measure and the generalization gap.

Figure 4: Change in the decision boundary according to the perturbation in parameter values $\theta \to \theta + \epsilon_{i,k}\Delta\theta_k$.

### 3.3.2 Connections to the margin

Our measure can also be linked to generalization in terms of margin. Unlike the usual definition of the margin as the distance from the decision boundary to its closest data, we devise a similar concept developed in the parametric model space rather than the input space.

Given a model parameter value, if we perturb the value so that the model's decision boundary passes through the location of a nearby data as shown in Figure 4, the difference between the initial and the perturbed parameter values would correspond to a sort of margin defined in the model parameter space. Since the parameter space dimension is high, we can perturb the model parameter value in various directions. Considering the perturbations of parameter values along the eigenvectors of the FIM results in an interesting connection between this margin and our sharpness measure. We now formalize this relationship for the binary classification case, while that for the multi-class case is derived in Appendix E.

We define the binary cross-entropy loss as $l_i(\theta) = l_{BCE}(x_i, z_i; \theta) = -z_i \log p(x_i; \theta) - (1 - z_i)\log(1 - p(x_i; \theta))$, where $z_i \in \{0, 1\}$ denotes the binary class label for an input $x_i \in \mathbb{R}^D$ and $p(x_i; \theta)$ denotes the model's prediction on the probability that $x_i$ belongs to class one. The condition for $x_i$ to lie on the decision boundary is $p(x_i) = \frac{1}{2}$, which is equal to $l_i(\theta) = \log 2$ regardless of the $z_i$ value.

Suppose we perturb the neural network parameter value from $\theta$ to $\theta + \epsilon_{i,k}\Delta\theta_k$, where $\Delta\theta_k \in \mathbb{R}^m$ is the $k$-th eigenvector of the FIM associated with the $k$-th eigenvalue $\lambda_k$ and $\epsilon_{i,k}$ is a scalar. For the decision boundary of the perturbed model to lie on $x_i$, the scalar $\epsilon_{i,k}$ should satisfy $\log 2 = l_i(\theta + \epsilon_{i,k}\Delta\theta_k) \approx l_i(\theta) + \epsilon_{i,k}\left(\frac{\partial l_i}{\partial \theta}\right)\Delta\theta_k$, where we apply the first-order Taylor expansion by assuming small $\epsilon_{i,k}$. The scalar $\epsilon_{i,k}$ can then be approximated as $\epsilon_{i,k} \approx \frac{\log 2 - l_i(\theta)}{\left(\frac{\partial l_i}{\partial \theta}\right)\Delta\theta_k}$.

A sort of margin can be derived in an information geometric sense by computing the squared norm of the perturbation vector $\epsilon_{i,k}\Delta\theta_k$ using the inner product based on the FIM as follows:

$$\epsilon_{i,k}^2 \Delta\theta_k^\top F(\theta)\Delta\theta_k = \epsilon_{i,k}^2 \lambda_k \approx \lambda_k \left(\frac{\log 2 - l_i(\theta)}{\left(\frac{\partial l_i}{\partial \theta}\right)\Delta\theta_k}\right)^2. \tag{10}$$

The summation over $k = 1, \ldots, m'$ (with $m'$ as the largest $k$ with non-zero $\lambda_k$) of the reciprocal of these squared norms (hence weighing more on small $\epsilon_{i,k}^2$ values which would reduce the error from approximations) can then be related to our sharpness measure as follows:

$$\sum_{k=1}^{m'} \frac{1}{\epsilon_{i,k}^2 \lambda_k} \approx \sum_{k=1}^{m'} \frac{1}{\lambda_k}\left(\frac{\left(\frac{\partial l_i}{\partial \theta}\right)\Delta\theta_k}{\log 2 - l_i(\theta)}\right)^2 = \frac{\left(\frac{\partial l_i}{\partial \theta}\right)F(\theta)^\dagger\left(\frac{\partial l_i}{\partial \theta}\right)^\top}{(\log 2 - l_i(\theta))^2}, \tag{11}$$

where we have used $F(\theta)^\dagger = \sum_{k=1}^{m'} \frac{1}{\lambda_k}\Delta\theta_k\Delta\theta_k^\top$ in deriving the last identity. Note that our sharpness measure at each data $(x_i, y_i)$ appears in the numerator of (11). This indicates that the smaller our sharpness measure, the larger the squared norms ($\epsilon_{i,k}^2 \lambda_k$) in the LHS, which means that the margin becomes larger in an information geometric sense.

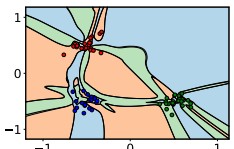
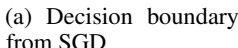
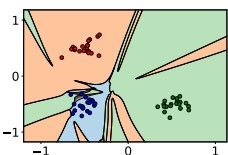
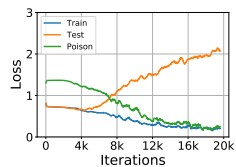
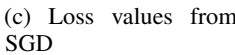
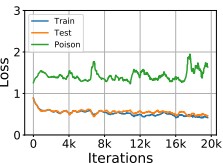

(a) Decision boundary from SGD

(b) Decision boundary from SGD with regularization

(c) Loss values from SGD

(d) Loss values from SGD with regularization

Figure 5: A synthetic three-class classification example trained using SGD with and without regularization under the effect of poison data. The loss values are smoothed for better visualization.

## 4 Using the sharpness measure as a regularizer to train neural networks

We now apply our sharpness measure as a regularizer to train neural networks to see if regularizing the measure can improve the generalization performance. The corresponding loss function is written as

$$\mathcal{L}(\theta) = \frac{1}{N} \sum_{i=1}^{N} l_i(\theta) + \rho \cdot \text{IGS}(\theta), \tag{12}$$

where $\rho$ is a coefficient for the regularization term. We consider a toy example and then consider image classification tasks involving MNIST and CIFAR-10/100 data sets. We also provide some tractable and possibly efficient ways to regularize our sharpness measure. All the experiments have been performed using the PyTorch library [38].

### 4.1 A toy example

We first apply our regularizer to a toy example. In [18], they have trained neural networks on poison data with the wrong labels to hamper the generalization performance and make the loss landscape extremely sharp while fitting all the training data. We follow this experimental setting and check whether the obtained solution can avoid the undesirable generalization performance and loss landscapes when applying our regularizer (calculated on the training data). The experimental details, as well as how the IGS is approximated for this example, are provided in Appendix G.3.

The experimental results are shown in Figure 5. The results from SGD without regularization show very irregular decision boundaries with tiny margins, whereas relatively soft boundaries with larger margins appear when our regularizer is used, especially around the training data. Applying our regularizer also shows a lower generalization gap, i.e., the difference between the training and test losses, than the SGD without regularization.

### 4.2 MNIST and CIFAR-10/100

To check the effect of the regularizer in more realistic settings, we apply our regularizer to the classification of MNIST and CIFAR-10/100 data sets. For this experiment, we regularize the mini-batch IGS defined in (7). For an efficient implementation of our regularizer, the following approximation on perturbed loss functions with a perturbation $\delta = \rho F(\theta)^{\dagger} \left( \frac{\partial l}{\partial \theta} \right)^{\top} \in \mathbb{R}^m$ (with a small $\rho > 0$) is useful:

$$l(\theta + \delta) \approx l(\theta) + \left( \frac{\partial l}{\partial \theta} \right) \delta \approx l(\theta) + \rho \cdot \left( \frac{\partial l}{\partial \theta} \right) F(\theta)^{\dagger} \left( \frac{\partial l}{\partial \theta} \right)^{\top}. \tag{13}$$

Note that for $l = \frac{1}{b} \sum_{x_i \in B} l_i$ with a mini-batch $B$ of size $|B| = b$, minimizing (13) becomes minimizing a stochastic version of (12) (with $\text{IGS}_b(\theta)$), where the stochasticity comes from sampling a mini-batch $B$. Therefore we minimize (13) in our experiments with $\rho$ as a tunable hyperparameter. To obtain $\delta$, we resort to an EKFAC-based approximation for the natural gradients [13] as detailed in Appendix G.4.1.

A three-layer fully connected neural network (3FCN) is used for the experiments using MNIST data, and various convolutional neural networks such as VGG [42], ResNet [15], and WideResNet [50] are used for CIFAR-10/100 experiments, with data augmentation, cosine annealing [27], and label

Table 1: Averages and standard errors of the test classification accuracies for SGD, GR, SAM, ASAM, and SGD with our regularization method on MNIST, CIFAR-10, and CIFAR-100 data sets.

| DATA SET | MODEL | SGD | GR | SAM | ASAM | OURS |
|---|---|---|---|---|---|---|
| MNIST | 3FCN | $98.13 \pm 0.05$ | $98.20 \pm 0.01$ | $98.62 \pm 0.01$ | $\mathbf{98.65} \pm 0.07$ | $\mathbf{98.65} \pm 0.03$ |
| CIFAR-10 | VGG11-BN | $92.75 \pm 0.11$ | $93.13 \pm 0.14$ | $93.61 \pm 0.08$ | $93.78 \pm 0.13$ | $\mathbf{93.81} \pm 0.06$ |
| | RESNET-56 | $94.08 \pm 0.17$ | $94.40 \pm 0.23$ | $95.22 \pm 0.11$ | $\mathbf{95.36} \pm 0.06$ | $94.96 \pm 0.01$ |
| | WRN-16-8 | $95.98 \pm 0.03$ | $96.15 \pm 0.10$ | $96.70 \pm 0.14$ | $\mathbf{97.03} \pm 0.06$ | $96.57 \pm 0.10$ |
| CIFAR-100 | VGG11-BN | $72.25 \pm 0.17$ | $73.05 \pm 0.32$ | $73.69 \pm 0.20$ | $73.61 \pm 0.18$ | $\mathbf{74.24} \pm 0.22$ |
| | RESNET-56 | $73.14 \pm 0.10$ | $74.67 \pm 0.25$ | $75.90 \pm 0.25$ | $75.87 \pm 0.27$ | $\mathbf{76.07} \pm 0.11$ |
| | WRN-16-8 | $80.55 \pm 0.33$ | $81.13 \pm 0.06$ | $82.20 \pm 0.11$ | $82.38 \pm 0.09$ | $\mathbf{82.44} \pm 0.29$ |

smoothing [30] methods additionally applied. For comparison, we consider the gradient regularization (GR) method [4, 43], sharpness-aware minimization (SAM) method [12], and the adaptive SAM (ASAM) method [24], an extension of SAM to involve the scale-invariance. The detailed experimental settings are explained in Appendix G.4.2.

We provide the averages and standard errors of the test classification accuracies obtained from three runs of each method in Table 1. As can be seen from the table, one can confirm that the generalization performance of SGD is significantly improved with our regularizer. Furthermore, our method shows better performance than the GR method and a comparable performance improvement with the SAM and ASAM methods.

## 5 Conclusion

Various empirical results observed from running deep learning algorithms such as SGD have suggested that minima with a flatter loss landscape tend to show better generalization performance. However, the previous definitions of sharpness/flatness have faced a severe dependence on reparametrizations or parameter scalings that do not change the model output and the generalization performance [8].

In this paper, based on an information geometric analysis of the neural network parameter space, we have proposed an information geometric sharpness measure which is reparametrization- and scale-invariant, making it free from the issues posed by [8]. We have also discussed the connection of the measure to generalization. In addition, a regularizer that reduces the suggested sharpness measure is proposed, showing a significant improvement in generalization performance under practical settings.

Unlike many other sharpness measures, the Fisher information matrix (FIM) plays a crucial role in our measure. Since it can be computationally demanding to calculate the FIM and our sharpness measure for large-scale neural networks, more efficient evaluation methods should be developed to apply our measure to such networks. Concerning the regularization of our measure during training, exploring better ways to reduce the time complexity and increase generalization performance is left for future work. Furthermore, analyzing the generalization properties of models obtained from deep learning algorithms such as SGD or natural gradient descent [2, 28] using our measure would be another intriguing future work.

## Acknowledgments and Disclosure of Funding

C. Jang and Y.-K. Noh were supported by IITP Artificial Intelligence Graduate School Program for Hanyang University funded by MSIT (Grant No. 2020-0-01373). S. Lee was supported by a KIAS Individual Grant (AP083601) via the Center for AI and Natural Sciences at Korea Institute for Advanced Study. F. C. Park was supported in part by SRRC NRF grant 2016R1A5A1938472, IITP-MSIT grant 2022-0-00480 (Training and Inference Methods for Goal-Oriented AI Agents), SNU-AIIS, SNU-IAMD, and the SNU Institute for Engineering Research. Y.-K. Noh was partly supported by NRF/MSIT (Grant No. 2018R1A5A7059549, 2021M3E5D2A01019545) and IITP/MSIT (Grant No. IITP-2021-0-02068).

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
