# OpenReview forum: "A Reparametrization-Invariant Sharpness Measure Based on Information Geometry"
_NeurIPS.cc/2022/Conference — NeurIPS 2022 Accept_

### Official Review · Reviewer_fVF7 · 2022-07-07

**Rating:** 7
**Confidence:** 4
**Soundness:** 4 excellent
**Presentation:** 3 good
**Contribution:** 3 good

**Summary:**

The paper investigates the spectrum of the Fisher Information Matrix (FIM) and empirically observes that the eigenspace of large eigenvalues is connected to large shifts of the decision boundary of the network. The authors then propose a novel sharpness measure that is based on the pseudo-inverse of the FIM and the loss gradients at each sample. In comparison to existing measures, the proposed measure is shown to be reparameterization-invariant (instead of just scale-invariant). A theoretical connection to generalization is provided by presenting the proposed measure as an approximation to Takeuchi’s information criterion, and showing that a measure of margin for a sample point can be formulated in terms of the proposed sharpness measure at this datapoint. Empirically, it is shown that using the sharpness measure as a regularization term during training, can improve generalization performance.



**Questions:**

Could the authors provide specifics in which sense they would argue their measure has a a theoretical connection to generalization that other sharpness measures are lacking?

What are the theoretical advantages over the TIC in terms of its connection to generalization?

Can the authors present evidence that their sharpness measure is experimentally advantageous to other measures?

Can the authors present a (weight) reparameterization of a neural network (with fixed architecture), that is not covered by scale invariance? In other words, are the authors aware of a (classical deep) neural network architecture that allows two distinct weight configurations, not in linear dependence, that lead to the same network function ?

**Limitations:**

As explained in detail above, I don't think that the limitations of the proposed measure are fairly discussed. According to the presentation, the proposed measure does not provide a stronger theoretical connection to generalization than competing measures, neither is it shown to outperform other measures experimentally.


**Strengths And Weaknesses:**

Strengths:

\+ The paper proposes a reparameterization-invariant sharpness measure. As competing measures are only shown to be scale-invariant, this is a clear improvement over other measures.  While this theoretical observation would be sufficient, it is backed up by a clear empirical result.

\+ The FIM is presented as a natural candidate for a measure due to its theoretical properties. In addition, by investigating the spectrum of the FIM, the authors empirically observe that the eigenspace of large eigenvalues is connected to large shifts of the decision boundary of the network.  This helps to naturally place the proposed measure into existing theory.

\+ Moreover, the paper illustrates interesting connection to the margin and the Takeuchi information criterion, which further offers indications that the proposed measure could be related to generalization.

\+ The empirical evaluation (on MINST, CIFAR10 and CIFAR100) supports this theoretical observations by showing that by using the sharpness measure as a regularization term during training, the performance of networks is improved.

\+ The paper is very well-written and easy to follow. The supplements add all the details left out in the main article.

Weaknesses:

\- The authors propose a novel sharpness measure with theoretical and empirical connections to generalization, but it remains unclear how the measure compares to competing measures. While the papers present their measure as largely superior in both theoretical and empirical properties, this is not a fair placement in related work:

(i) Theory: The authors write that for sharpness measures,"any theoretical insights into generalization performance“ (Line 5) are "still lacking“ and that they address other measures “inability to explain generalization performance of deep learning networks“ (Line 39). However, other sharpness measures have been theoretically connected to generalization, for example in terms of generalization bounds. At the same time, the presented connection to generalization for the proposed measure is not that strong. Firstly, the proposed measure is an approximation to another measure (TIC) that has been connected to correlation empirically. If the authors aim to borrow the theoretical properties of the TIC for their connection to generalization, then the TIC itself (i.e. replace the FIM with the Hessian in the proposed measure) would be the superior measure (a measure that has been discussed in related work). Secondly, the connection to the margin requires a (possibly tailored) novel definition of margin .

(ii) Experiments: The experiments successfully show that the proposed measure can be employed as a regularization term to improve generalization. However, the experiments do neither show that this approach outperforms existing approaches (SAM, ASAM), nor do they show that the proposed measure outperforms other sharpness measures. Other sharpness measures have been similarly applied to improve generalization, but no comparison is provided.  There is also no comparison of different sharpness measures in their ability to explain generalization. In particular, it would be important to compare to the TIC as used in the reference [44] to the proposed measure.

(iii) Reparameterization-invariance: A clear improvement of the proposed measure over other sharpness measures is its reparameterization-invariance, which goes beyong scale-invariance. However, the benefit may not be as clear as it first seems. The reparameterizations that are considered as evidence for the failing of other measures, are not reparameterizations of the weights in a network, but they go beyond the (classical) setting of neural networks. So, to my knowledge, the linear reparameterizations (that competing measures are invariant to) may be the only reparameterizations that play a role in the setting of neural networks.

\- The motivation for introducing the novel sharpness measure is not well-explained. It seems plausible that the Takeuchi’s information criterion (TIC) and/or the calculations on the margin in 3.3.2 motivated the definition of the novel measure, but the paper presents no explicit motivation for the introduction of the measure ("Based on the above analysis" in line 169 does not point to anything that explains the explicit form of the measure)  and then presents a relation to the TIC and the margin.

*****

Taken together, the paper offers an interesting addition to the set of sharpness measures and their ability to explain generalization. However, the paper does not present clear evidence that the proposed measure is superior to other sharpness measures empirically or theoretically. Unless advantages and motivation could be worked out more clearly, the paper should not claim its superiority over related work.

-----
Typos:

25: masures

242: Possibly revise this statement, which says that "When we perform the maximum likelihood estimation [..], we can obtain the maximum likelihood estimate [..]"

---

> ### Author Response · Authors · 2022-08-01
> **Reply to reviewer's comments #1**
>
> ### About weaknesses
> * Regarding the mentioned weaknesses in the comparison with other measures, since the reviewer asks similar questions in the **Questions**, please refer to our **Answers to questions** below.
> * As pointed out by the reviewer, the connections with generalization (via TIC and margin) motivated our proposal of the specific form of measure in (6). (This is currently explained in manuscript lines 184-186.) To further reveal this motivation, we will consider replacing the 'Based on the above analysis,' with 'To reflect the above analysis and to connect the sharpness measure to generalization (as we explain later),'.
> * We appreciate the reviewer for the careful reading. We will fix the typo.
>
> ### Answers to questions
> * We would like to clarify that our measure can indeed be mathematically connected to the generalization gap. TIC, which our measure can be approximated to, goes beyond the empirical correlation with generalization. \
> Under asymptotic assumptions, TIC theoretically estimates the bias $b(\hat{\theta})$ of the maximum log-likelihood value, i.e., TIC = $\lim\limits_{N \to \infty} b(\hat{\theta})$ with $N$ as the number of training data. The bias is defined as follows:\
> $ b(\hat{\theta}) = E_{\mathcal{D}}\left[ \frac{1}{\left|\mathcal{D}\right|}\sum\limits_{(x_i, y_i) \in \mathcal{D}}\log p(x_i, y_i; \hat{\theta}(\mathcal{D})) – E_{(x,y)}\left[ \log p(x, y; \hat{\theta}(\mathcal{D}))\right]\right]$, where $\mathcal{D}$ = $\\{ (x_1, y_1), \ldots, (x_N, y_N) \\}$ is the training data with $(x_i, y_i) \sim p(x,y)$ and $\hat{\theta}(\mathcal{D})$ denotes the maximum likelihood estimate obtained from using $\mathcal{D}$ [1'].
> Since our loss is the negative log-likelihood, the first and second term inside $E_{\mathcal{D}}[\cdot]$ corresponds to the (negative) training and test losses, respectively. Consequently, it can be said that TIC estimates the expected generalization gap (i.e., test loss – training loss) of the parametric model. \
> We will revise Section 3.3.1 of the main text to emphasize these theoretical aspects further. \
> Among the existing sharpness measures, there are some measures that lack any connections to generalization (e.g., those in [9',10',11']), and other measures that can be connected to the generalization bound (e.g., [12',13']). To fairly place our work in the context of the existing literature, we will consider revising the following expressions:\
> (lines 5-6) ‘(...) lacking, as are any theoretical insights into generalization performance.’ -> ‘(...) lacking. Moreover, they often do not provide any theoretical insights into generalization performance.’\
> (line 39) ‘their inability to explain generalization performance of deep learning networks.’ -> ‘the problem that they often do not explain generalization performance of deep learning networks.’
>
> * It is difficult to characterize the theoretical advantages of our measure over TIC in terms of its link to generalization. However, our measure can be approximated to the TIC after sufficient training of the model. (Figure 5 (left) of [5'] shows experimental results for replacing the Hessian with the FIM, which exactly corresponds to our approximation.) Also, since our measure is defined based on the FIM, reparametrization-invariance is guaranteed theoretically, and empirically the calculation is much easier than the TIC that includes the Hessian.
>
> [1'] Dixon, M., Ward, T.: Takeuchi’s information criteria as a form of regularization. arXiv preprint arXiv:1803.04947 (2018)
>
> [5'] Thomas, V., Pedregosa, F., Merriënboer, B., Manzagol, P.A., Bengio, Y., Le Roux, N.: On the interplay between noise and curvature and its effect on optimization and generalization. In: International Conference on Artificial Intelligence and Statistics, pp. 3503–3513. PMLR (2020)
>
> [9'] Keskar, N.S., Nocedal, J., Tang, P.T.P., Mudigere, D., Smelyanskiy, M.: On large-batch training for deep learning: Generalization gap and sharp minima. In: 5th International Conference on Learning Representations, ICLR 2017 (2017)
>
> [10'] Yao, Z., Gholami, A., Lei, Q., Keutzer, K., Mahoney, M.W.: Hessian-based analysis of large batch training and robustness to adversaries. Advances in Neural Information Processing Systems 31 (2018)
>
> [11'] Rangamani, A., Nguyen, N.H., Kumar, A., Phan, D., Chin, S.P., Tran, T.D.: A scale invariant measure of flatness for deep network minima. In: ICASSP 2021-2021 IEEE International Conference on Acoustics, Speech and Signal Processing (ICASSP), pp. 1680–1684. IEEE (2021)
>
> [12'] Petzka, H., Kamp, M., Adilova, L., Sminchisescu, C., Boley, M.: Relative flatness and generalization. Advances in Neural Information Processing Systems 34 (2021)
>
> [13'] Tsuzuku, Y., Sato, I., Sugiyama, M.: Normalized flat minima: Exploring scale invariant definition of flat minima for neural networks using pac-bayesian analysis. In: International Conference on Machine Learning, pp. 9636–9647. PMLR (2020)

---

> > ### Comment · Reviewer_fVF7 · 2022-08-09
> > **Reply to authors response**
> >
> > I thank the authors for their clarifying comments.
> >
> > In particular, I appreciate the specific examples of neural networks with quadratic activations, and I agree with the comment that it is important to construct "more fundamental measures by considering such reparametrizations". The invariance to such nonlinear reparameterizations indeed seems like an improvement, although it may not cover additional reparameterizations to scale-invariance for the specific setting of ReLU neural networks.
> >
> > I also appreciate the clarification on experimental results comparing competing measures.
> >
> > Although I am still not convinced that the theoretical connection to generalization (based on showing it to be an approximation to the TIC) is more convincing than for some competing measures, I increase my score to "7- Accept" as a result of the author's response.

---

> > > ### Author Response · Authors · 2022-08-09
> > > **Reply to reviewer's comments**
> > >
> > > We appreciate the reviewer for asking interesting questions which have helped to improve our manuscript.
> > >
> > > We also appreciate the reviewer for the kind replies and for raising the score.

---

> ### Author Response · Authors · 2022-08-02
> **Reply to reviewer's comments #2**
>
> ### Answers to questions
> * Experimental advantages over other measures may include:\
> (1) When compared with other sharpness measures to examine the indicative ability of the generalization gap, our measure has shown better correlations. In the MNIST experiment, the correlation coefficients to the generalization gap for each measure are [our measure: 0.661, $\text{Tr}(H)$: 0.608, FR norm [14']: 0.366, Rang et al. [11']: 0.565, Petzka et al. [12']: 0.570]. In the CIFAR-10 experiment, the coefficients are [our measure: 0.612, $\text{Tr}(H)$: 0.546, FR norm [14']: 0.233, Rang et al. [11']: 0.538, Petzka et al. [12']: 0.555]. (When calculating our measure, we consider the expectation taken with respect to the data distribution (similarly to [5']), which is approximated by a finite sum of the integrands over the test data.)\
> (2) Other sharpness measures that we compare in the above experiments, e.g., those of Liang et al. [14'], Rangamani et al. [11'], Petzka et al. [12'], have not been used as regularizers to improve the generalization performance during the model training. In particular, for measures that contain the Hessian (e.g., those of Rangamani et al. and Petzka et al.), implementing a regularizer that reduces the measure may not be straight-forward.
>
> * We present below an example of modeling the same network function within the same architecture but with two distinct weight configurations in nonlinear dependence.
> Consider $s(x) = x^2$ as the nonlinear activation function to apply. Denote by $ W^{(1)} \in \mathbb{R}^{3\times 2}$ the input-to-hidden weight and by $ W^{(2)} \in \mathbb{R}^{1\times 3}$ the hidden-to-output weight. The network function then becomes \
> $ f = \sum_{i=1}^3 W_{1i}^{(2)} (W_{i1}^{(1)} x_1 + W_{i2}^{(1)} x_2)^2$, where $ W_{ij}$ is the $(i,j)$ entry of $W$. \
> Set another network with the same architecture as \
> $\tilde{f} = \sum_{i=1}^3 V_{1i}^{(2)} (V_{i1}^{(1)} x_1 + V_{i2}^{(1)} x_2)^2$ using weights $V^{(1)} \in \mathbb{R}^{3\times 2}, V^{(2)} \in \mathbb{R}^{1\times 3}$. \
> The condition for the two networks modeling identical functions is as follows: \
> $\sum_{i=1}^3 W_{1i}^{(2)} (W_{i1}^{(1)} x_1 + W_{i2}^{(1)} x_2)^2 = \sum_{i=1}^3 V_{1i}^{(2)} (V_{i1}^{(1)} x_1 + V_{i2}^{(1)} x_2)^2$ for all $x \in \mathbb{R}^2$. \
> If we rearrange the above for each coefficient of $ x_1^2, x_1 x_2$, and $x_2^2$,
> we obtain \
> (i) $\sum_{i=1}^3 W_{1i}^{(2)} (W_{i1}^{(1)})^2 = \sum_{i=1}^3 V_{1i}^{(2)} (V_{i1}^{(1)})^2,$ \
> (ii) $\sum_{i=1}^3 W_{1i}^{(2)} W_{i1}^{(1)} W_{i2}^{(1)} = \sum_{i=1}^3 V_{1i}^{(2)} V_{i1}^{(1)} V_{i2}^{(1)},$ and \
> (iii) $\sum_{i=1}^3 W_{1i}^{(2)} (W_{i2}^{(1)})^2 = \sum_{i=1}^3 V_{1i}^{(2)} (V_{i2}^{(1)})^2$. \
> For the set of $ W^{(1)}, W^{(2)}, V^{(1)}, V^{(2)}$ that satisfy above equations, the two network functions become identical. \
> Consider finding $V^{(2)}$ for given $ W^{(1)}, W^{(2)}$, and $V^{(1)} = \phi_1(W^{(1)})$ for a bijective nonlinear function $\phi: \mathbb{R}^{3\times 2} \rightarrow \mathbb{R}^{3\times 2}$. Since there are three equations (i,ii,iii) and three variables of $(V_{11}^{(2)}, V_{12}^{(2)}, V_{13}^{(2)})$ in $V^{(2)}$, we can find the solution of $V^{(2)}$ and represent it as $V^{(2)} = \phi_2(W^{(1)}, W^{(2)})$. \
> We can then figure out that two different weight configuration with a nonlinear relationship between each other as  $V^{(1)} = \phi_1(W^{(1)})$ and $V^{(2)} = \phi_2(W^{(1)}, W^{(2)})$ can model identical network functions. \
> When the input or output dimension increases, if the hidden variable dimension increases appropriately, we can find mappings between weights that can model identical neural networks similarly. \
> Apart from the above example, even though general nonlinear reparametrizations may not fit perfectly in the typical setting of neural networks, we would like to insist that it can help devise interesting and even more fundamental measures by considering such reparametrizations. \
> In addition, for scale-invariant measures, when the activation function does not satisfy the homogeneity condition of the ReLU, so the scale-invariance disappears, the motivation to apply the measure may be significantly lost. On the other hand, for our measure, the meaning of measuring sharpness for a unit change of the probability density model can be maintained regardless of which activation function is used.
>
> ### About limitations
> Please refer to the above **Answers to questions**.
>
> [14'] Liang, T., Poggio, T., Rakhlin, A., Stokes, J.: Fisher-rao metric, geometry, and complexity of neural networks. In: The 22nd International Conference on Artificial Intelligence and Statistics, pp. 888–896. PMLR (2019)

---

### Official Review · Reviewer_uwH8 · 2022-07-08

**Rating:** 6
**Confidence:** 4
**Soundness:** 3 good
**Presentation:** 3 good
**Contribution:** 2 fair

**Summary:**

The article proposes a novel flatness measure for probabilistic classification models, parametrized as neural networks. The measure is computed as combination of inverse Fisher information matrix and first order derivatives of the loss function (formula (6)). The measure is justified from the information geometry perspective, where FIM allows to measure the difference between distributions. The other parts of the flatness measure are justified by introducing a similarity to the Takeuchi's information criterion, that nicely correlates with generalization. The measure is provably indifferent to a general reparametrization, when the parameters of the model are mapped to another set of parameters, as compared to other proposed measures, that are invariant only to scale reparametrizations. It is empirically demonstrated that introducing the proposed measure as regularization to the training procedure improves generalization.

**Questions:**

1 - Is there a significant difference between the properties of the Hessian and FIM: the discussion in the introduction attracts attention to the different directions that contribute more or less to the change of the considered function, but it is as well property of the Hessian?
2 - It seems that any modification of the parameters will not be changing the proposed flatness measure due to its form. Is it so and does it mean that even for different models the measure can be the same?

----
I thank the authors for the detailed answers in the rebuttal phase. Nevertheless, I still have my doubts about the reparametrizations used and the consequences of being insensitive to them, therefore I stay with my initial score.

**Limitations:**

It would be beneficial to address the limitations of the theoretical connection to generalization, in a sense of similarity to empirically proven indicators of generalization. Also, more detailed analysis of the limitations of the computation of the proposed measure would be in place.

**Strengths And Weaknesses:**

The proposed measure addresses the general reparametrization problem, that is not restricted to the scaling reparametrizations of the ReLU networks. It is provably invariant and helps the training process when used as regularizer.
The idea is presented very clear with sufficient explanations and good reproducibility of the experiments.
Nevertheless, it is hard to pinpoint how exactly this flatness measure improves the understanding of the training procedure and helps to create a link to generalization. Different from Petzka et al. this work does not mathematically represent the test loss or generalization gap in terms of the proposed flatness measure, only the indicative connection to TIC and max-margin are provided. In the empirical sense, as a regularization, the measure seems to perform on par with other flatness based regularizers as SAM and ASAM (table2 in appendix).
The correlation with generalization is reported only in appendix and it is not clear if the proposed flatness can be used as an indicator of good generalization, or it is only improving the training process.

Important to the originality of the proposed measure is the definition of reparametrizations. It would be good to provide it in the introduction already - it is very common in the literature to call scalings as reparametrizations, while explicit bijection of parameter space is not so common. Also, it seems to be hard to define it: Dinh et al. paper is referenced for an example of the bijection, but a reparametrization proposed there in the Section5 is batch normalization. The reparametrization demonstrated in the paper experiments is most probably taken from the appendix of the Dinh et al. paper and it has modified formulas - I would find it very useful if it is demonstrated that this bijection is indeed not affecting the network output (and loss) and also more attention is put on the explanation and variety of such reparametrizations.

It is mentioned in the paper that FIM is hard to compute, especially for large models, but it was not quite clear explained how this was addressed in the experiments: 4.1 does not contain the way it was computed, 4.2 proposes approximation to the gradient, and the generalization correlation contains another approach.

Minor:
Figure 4 would benefit from the explanation of why (a) shows less regular boundaries than (b), because they look somehow similar.

---

> ### Author Response · Authors · 2022-08-01
> **Reply to reviewer's comments #1**
>
> ### About weaknesses
> * We would like to clarify that our measure can indeed be mathematically connected to the generalization gap. TIC, which our measure can be approximated to, goes beyond the empirical correlation with generalization.
> Under asymptotic assumptions, TIC theoretically estimates the bias $b(\hat{\theta})$ of the maximum log-likelihood value, i.e., TIC = $\lim\limits_{N \to \infty} b(\hat{\theta})$ with $N$ as the number of training data. The bias is defined as follows:\
> $ b(\hat{\theta}) = E_{\mathcal{D}}\left[ \frac{1}{\left|\mathcal{D}\right|}\sum\limits_{(x_i, y_i) \in \mathcal{D}}\log p(x_i, y_i; \hat{\theta}(\mathcal{D})) – E_{(x,y)}\left[ \log p(x, y; \hat{\theta}(\mathcal{D}))\right]\right]$, where $\mathcal{D}$ = $\\{ (x_1, y_1), \ldots, (x_N, y_N) \\}$ is the training data with $(x_i, y_i) \sim p(x,y)$ and $\hat{\theta}(\mathcal{D})$ denotes the maximum likelihood estimate obtained from using $\mathcal{D}$ [1'].
> Since our loss is the negative log-likelihood, the first and second term inside $E_{\mathcal{D}}[\cdot]$ corresponds to the (negative) training and test losses, respectively. Consequently, it can be said that TIC estimates the expected generalization gap (i.e., test loss – training loss) of the parametric model. \
> We will revise Section 3.3.1 of the main text to further emphasize these theoretical aspects.
> * When compared with other sharpness measures to examine the indicative ability of the generalization gap, our measure has shown better correlations. In the MNIST experiment, the correlation coefficients to the generalization gap for each measure are [our measure: 0.661, tr(H): 0.608, FR norm: 0.366, Rang et al.: 0.565, Petzka et al.: 0.570]. In the CIFAR-10 experiment, the coefficients are [our measure: 0.612, tr(H): 0.546, FR norm: 0.233, Rang et al.: 0.538, Petzka et al.: 0.555]. (When calculating our measure, we consider the expectation taken with respect to the data distribution (similarly to [5']), which is approximated by a finite sum of the integrands over the test data.)
> * We will revise the introduction to include the definition of reparametrization that we mainly consider, which also appears at the beginning of Section 5 of Dinh et al. [6']. The reparametrization we considered in our experiments is the one in Figure 5 of Dinh et al. [6']. These bijections do not affect the network output by definition, and can be of infinitely many types (e.g., consider linear transform $\theta’ = \alpha*\theta$, with $\alpha \in \mathbb{R}$).
> * We appreciate the reviewer for pointing out the missing details about calculating the FIM in the text. In our experiments, several approximations have been used for the eigendecomposition (required to get the pseudo-inverse) of the FIM. \
> Section 4.1 and Appendix E are based on power iterations. The experiments in Section 4.1 utilized the functions of the pyHessian library [7']. In implementing the power iteration in Appendix E, we handle matrices of size [mini-batch data size $\times$ parameter size]. Since memory usage and computation costs will be proportional to the network size, additional approximations may be needed to handle larger neural networks. \
> Section 4.2 is based on the EKFAC method [3']. This approximates the FIM as a layer-wise block-diagonal matrix and efficiently performs eigendecomposition via Kronecker factorization for each block. This method may be more inaccurate than the power iteration, but it can be applied to larger neural networks. \
> The missing details and limitations of the FIM approximation used in the paper will be made clearer in the paper.
> * About Figure 4: observing the interval between each data point and the decision boundary, that of (a) is very close to the decision boundary for most data points when compared to that of (b) due to the complex shape of the decision boundary.
>
>
> [1'] Dixon, M., Ward, T.: Takeuchi’s information criteria as a form of regularization. arXiv preprint arXiv:1803.04947 (2018)
>
> [3'] George, T., Laurent, C., Bouthillier, X., Ballas, N., Vincent, P.: Fast approximate natural gradient descent in a kronecker factored eigenbasis. Advances in Neural Information Processing Systems 31 (2018)
>
> [5'] Thomas, V., Pedregosa, F., Merriënboer, B., Manzagol, P.A., Bengio, Y., Le Roux, N.: On the interplay between noise and curvature and its effect on optimization and generalization. In: International Conference on Artificial Intelligence and Statistics, pp. 3503–3513. PMLR (2020)
>
> [6'] Dinh, L., Pascanu, R., Bengio, S., Bengio, Y.: Sharp minima can generalize for deep nets. In: International Conference on Machine Learning, pp. 1019–1028. PMLR (2017)
>
> [7'] Yao, Z., Gholami, A., Keutzer, K., Mahoney, M.W.: Pyhessian: Neural networks through the lens of the hessian. In: 2020 IEEE International Conference on Big Data (Big Data), pp. 581–590. IEEE (2020)

---

> > ### Comment · Reviewer_uwH8 · 2022-08-02
> > **Reply**
> >
> > 1 - If TIC approximates the gap with the infinite amount of examples for likelihood, it means that the proposed flatness measure can be theoretically linked with generalization gap only with crossentropy loss?
> >
> > 2 - I believe linear transformation you mention (just multiplication by a scalar) actually does change the output of the function. I see an example of non-linear transformation with quadratic activations in other replies here, but can you maybe explain how the transformation that you applied for the experiments act and why does not it change the output?
> >
> > 3 - The computations limitations would also be interesting to discuss in more details from the perspective of what is actually computed in the regularization term - would it be just (approximate) Hessian then?
> >
> > 4 - So by regularity of the bounds you mean more the margin? I think the caption would be clearer if you write it this way.

---

> > > ### Author Response · Authors · 2022-08-09
> > > **Reply to reviewer's additional questions**
> > >
> > > We appreciate the reviewer for the kind reply. \
> > > \
> > > 1 – Since we do not assume any specific form of the likelihood during the derivation of TIC, the TIC and our flatness measure (under the condition that the Hessian can be approximated to the FIM) can be linked to other losses according to the choice of the likelihood. For example, our measure can be defined for the square loss by considering the log-likelihood for a gaussian. \
> > > \
> > > 2 – In our experiments, the reparametrized models are defined as follows (please also refer to line 230 of the manuscript): For $\theta \mapsto \eta = g(\theta)$, the reparametrized model $\tilde{f}(\cdot; \eta) \equiv f(\cdot; g^{-1}(\eta)) = f(\cdot; \theta)$. The computation regarding the reparametrized model is done according to the above definition. That is, if the parameter $\eta$ for $\tilde{f}$ is given, we find corresponding $\theta = g^{-1}(\eta)$ and apply the obtained $\theta$ to the original model $f$. Therefore, $\tilde{f}(x;\eta) = f(x;\theta)$ is always satisfied. (Note that the sharpness values can differ between the two models since we measure the sharpness with respect to $\eta$ for the reparametrized model and with respect to $\theta$ for the original model.)\
> > > For the case of the mentioned linear map $\eta = g(\theta) = \alpha * \theta$, if we set $f(x; \theta) = x^\top \theta$ as an example, the corresponding reparametrized model is obtained by the above definition as $\tilde{f}(x; \eta) = f(x; \frac{1}{\alpha} \eta) = \frac{1}{\alpha} x^\top \eta = x^\top \theta$. Therefore, the model output remains the same between the original and reparametrized models for all $x$. Under the identical logic, the reparametrization $\eta = g(\theta) = (|\theta - \hat{\theta}|^2 + b)^a (\theta - \hat{\theta}) + \hat{\eta}$ that we used in the experiments does not change the model output.\
> > > In general, the form of $\tilde{f}$ (with respect to $\eta$) would differ from that of $f$ (with respect to $\theta$). (From the above example, $\tilde{f}(x; \eta) = \frac{1}{\alpha} x^\top \eta$ and $f(x;\theta) = x^\top \theta$.) But the example of non-linear transform with quadratic activations in other replies (for the reviewers q6RN and fVF7) corresponds to a special case that satisfies $f(x; \theta) = f(x; g(\theta)) = f(x;\eta) $ hence $f(x;\eta) = \tilde{f}(x;\eta)$ for all $x$.\
> > > \
> > > 3 – For an efficient implementation of the regularizers, in Section 4.1, we utilize the pyHessian library to obtain the top few ($m'=3$) eigenvectors/eigenvalues of the Hessian, i.e., $\Delta \theta_k$ and $\lambda_k$ for $k=1,\ldots,m'$ in line 636 (Here we used that the Hessian can be approximated to the FIM). We then obtain a perturbation vector $v_\epsilon = \sum_{k=1}^{m'} \frac{\epsilon_k}{\sqrt{\lambda_k+\eta}} \Delta\theta_k$ and use the loss values from the perturbed parameters as in Eq (39) to approximate the regularization term. \
> > > In Section 4.2, we need to obtain $\delta = \rho F(\theta)^\dagger \left(\frac{\partial l}{\partial \theta}\right)^\top$ in Eq (12). According to the layer-wise block-diagonal assumption for the FIM, the eigendecomposition is done block-wisely based on the EKFAC method, e.g., for the $i$-th block $F(\theta)_{i} = U_i S_i U_i^\top$,
> > >
> > > where $U_i, S_i$ are the eigenvector and eigenvalue matrices. The pseudo-inverse of the FIM can be obtained block-wisely as $F(\theta)^\dagger_{i} = U_i S_i^{-1} U_i^\top$. The multiplication of $F(\theta)^\dagger$ with the gradient $\left(\frac{\partial l}{\partial \theta}\right)^\top$ can also be done block-wisely. Note that we do not need to obtain ${F( \theta )}_{i}$
> > >
> > > or $F( \theta )^\dagger_{i}$ explicitly, but only the multiplication $\delta = \rho F(\theta)^\dagger \left(\frac{\partial l}{\partial \theta}\right)^\top$. Therefore, the actual computation performed after obtaining $U_i, S_i$ is $\delta_i = \rho U_i S_i^{-1} U_i^\top \left(\frac{\partial l}{\partial \theta}\right)_i^\top$ for $i = 1, \ldots, L$, where $L$ is the number of blocks and $\delta_i, \left(\frac{\partial l}{\partial \theta}\right)_i^\top$ are the respective components of $\delta, \left(\frac{\partial l}{\partial \theta}\right)^\top$ corresponding to the $i$-th block. \
> > > \
> > > 4 – Yes. We appreciate the reviewer’s suggestion and will revise the sentences accordingly.

---

> ### Author Response · Authors · 2022-08-01
> **Reply to reviewer's comments #2**
>
> ### Answers to questions
> 1. The Hessian has similar properties. To see why, consider the following Hessian decomposition  [8']:
>
> $H = \frac{1}{N} \sum_{i=1}^N \left( J_i^\top (diag(p_i) – p_i p_i^\top) J_i + \sum_{k=1}^C ( p_i – e_{y_i})_k \frac{\partial^2 f_k}{\partial \theta^2} \right).$
>
> When the neural network is sufficiently trained, the latter term can become very small (i.e., $p_i – e_{y_i} \approx 0$), so the FIM (corresponding to the first term) is not significantly different from the Hessian. Therefore, we can say that there are no significant differences between the two matrices in terms of having a few outliers and a bulk near zero in the eigenspectra [8'].
>
> 2. As the reviewer mentioned, the form of IGS does not need to be changed for different network models.
>
> ### About limitations
> Please refer to the first bullet and the second last bullet in **About weaknesses**.
>
> [8'] Papyan, V.: Measurements of three-level hierarchical structure in the outliers in the spectrum of deepnet hessians. In: International Conference on Machine Learning, pp. 5012–5021. PMLR (2019)

---

> > ### Comment · Reviewer_uwH8 · 2022-08-02
> > **Reply**
> >
> > 1 - My problem with similarity of Hessian to FIM is that it means that the argumentation line in the introduction of the paper is not quite justified: FIM in itself does not provide some new insightful information about the loss surface, because the insights are similar to the ones got from Hessians. It means that FIM-based flatness measure can be just yet another measure, that is not sensitive to additional class of reparametrizations.
> >
> > 2 - I think my question was not formulated clear: what I mean is that will not the measure be insensitive to _any_ modification of the parameters, even the one that does lead to the change of the output of the model?

---

> > > ### Author Response · Authors · 2022-08-09
> > > **Reply to reviewer's additional questions**
> > >
> > > 1 – The insights obtained from the eigenspectra of the FIM are indeed similar to those from the Hessian in the sense that the amount of the parametric model change or the loss change varies a lot according to the change direction in parameter values. However, such a concept has not been applied much in defining existing sharpness measures. \
> > > Previously existing hessian-based measures have weighed all parameters identically and have considered the sharpness along the sharpest direction (the top eigenvalue) or the average sharpness (the trace) with respect to the unit change of parameter values. On the other hand, we measure the sharpness with respect to the unit change in the parametric model, which naturally reflects the different influences on the parametric model according to the parameter change directions. Interestingly, implementing this idea results in using the ‘pseudo-inverse’ of the FIM to define the sharpness, quite a different way of using the Hessian from existing Hessian-based measures.\
> > > Our sharpness measure has distinguishable features from other measures since it thoroughly reflects the insight from the eigenspectra of the FIM, possesses both the reparametrization- and scale-invariance, and can also be connected well to the generalization both theoretically and empirically.\
> > > \
> > > 2 – We apologize for misinterpreting the reviewer’s initial question. Our measure is indeed sensitive to the change in the parameter values. If parameter values change, both $F(\theta)$ and $\frac{\partial l_i}{\partial \theta}$ values can change, and also our sharpness measure which is obtained by multiplying these components. Note that the invariance we have shown in Proposition 1 is limited to the transforms satisfying the assumptions in the proposition. If we apply parameter transform that do not satisfy the assumption, the conditions $\left(\frac{\partial f}{\partial \theta}\right) = \left(\frac{\partial f}{\partial g(\theta)}\right) \left(\frac{\partial g(\theta)}{\partial \theta}\right)$ and $F(\theta) = \left(\frac{\partial g(\theta)}{\partial \theta}\right)^\top F(g(\theta)) \left(\frac{\partial g(\theta)}{\partial \theta}\right)$ (these are terms in Appendix D required to prove Proposition 1) will not hold in general hence Eq. (28) will not hold also and can have different sharpness values. We can also figure this out experimentally on the y-axis of the scatter plots in Figure 5, which show the sharpness measure values obtained from different parameter values.

---

### Official Review · Reviewer_q6RN · 2022-07-10

**Rating:** 6
**Confidence:** 3
**Soundness:** 4 excellent
**Presentation:** 3 good
**Contribution:** 3 good

**Summary:**

The paper develops a new sharpness measure based on Fisher information. The proposed measure is invariant to scaling and reparametrization and is connected to generalization performance. Experiments shows that using the proposed sharpness measure as a regularizer significantly improves generalization for SGD, and achieves comparable performance as other sharpness-aware methods.

**Questions:**

- In addition to the time reported at the end of the appendix, is it possible to provide more details about the added time complexity of using the (approximated) Fisher information matrix?
- Proposition 3.1 seems to be more general than scaling invariance. In different architectures, are there other mappings other than scaling that satisfy the assumptions in the proposition?


**Limitations:**

The authors have addressed limitations in the conclusion.


**Strengths And Weaknesses:**

Strengths:
- The use of Fisher information in developing a scale- and reparametrization- invariant sharpness measure is novel and well motivated.
- The paper is technically sound. The authors present the background and mathematical details in a way that is easy to read.
- The connections to existing concepts in generalization are interesting and insightful.

Weaknesses:
- From Table 1, it is not clear that regularization using the proposed sharpness measure can outperform other sharpness-aware methods. Additionally, the proposed method is slower than SAM and ASAM, which makes its empirical advantage unclear.

---

> ### Author Response · Authors · 2022-08-01
> **Reply to reviewer's comments #1**
>
> ### About weaknesses
> * Although the empirical advantage of regularizing our measure over other sharpness-aware minimization techniques may be somewhat unclear, the main purpose of the experiments was to verify our measure's applicability as a regularizer. Based on our measures, exploring further ways to reduce time complexity and increase generalization performance (e.g., using other efficient methods of approximating the Fisher information matrix) is left for future work.
>
> ### Answers to questions
> * In our regularization method, the process of finding the natural gradient is added once every iteration compared to vanilla SGD. Therefore, additional time complexity depends on the details of methods to obtain the natural gradients. For example, In the case of the EKFAC method [3'] we use, the time complexity becomes the addition of that for computing gradient and that for solving layer-wise eigenvalue problems. Due to the Kronecker factorization, the time complexity of the latter per layer becomes $O(d^3)$, where $d$ denotes the typical layer input and output node size. (In EKFAC, the eigenvalue problem is usually solved per several tens or hundreds of iteration steps.) The final complexity is $O(m l d^2)$ for vanilla SGD, and $O(m l d^2) + O(l d^3)$ for the EKFAC, where $l$ is the number of layers, $m$ is the mini-batch size. (For a more detailed analysis of the coefficients of each term, please refer to Section 8 of the KFAC paper [4'].) We appreciate the reviewer for making such a useful question, and we will summarize this discussion in the appendix.
>
>
> [3'] George, T., Laurent, C., Bouthillier, X., Ballas, N., Vincent, P.: Fast approximate natural gradient descent in a kronecker factored eigenbasis. Advances in Neural Information Processing Systems 31 (2018)
>
>
> [4'] Martens, James, and Roger Grosse. "Optimizing Neural Networks with Kronecker-factored Approximate Curvature." arXiv preprint arXiv:1503.05671 (2015).

---

> ### Author Response · Authors · 2022-08-01
> **Reply to reviewer's comments #2**
>
> ### Answers to questions
> * We can make several examples of transforms that satisfy the assumption of Proposition 3.1.
>   * For the fully-connected neural networks, we can apply identical permutations for the $i$-th layer weights row-wisely and the $i+1$-th layer weights column-wisely. (For CNNs, applying permutations to channels would be similar.)
>   * For the fully-connected neural networks, if we apply odd functions (i.e., $\phi(-u) = -\phi(u)$) as an activation function, we can apply sign changes to some rows of the $i$-th layer weight and corresponding columns of the $i+1$-th layer weights.
>   * As a bit more complicated nonlinear example, consider a neural network with two-dimensional input $ x = (x_1, x_2)$, three-dimensional hidden layer, and one-dimensional output. Consider $s(x) = x^2$ as the nonlinear activation function to apply. Denote by $ W^{(1)} \in \mathbb{R}^{3\times 2}$ the input-to-hidden weight and by $ W^{(2)} \in \mathbb{R}^{1\times 3}$ the hidden-to-output weight. The network function then becomes \
> $ f = \sum_{i=1}^3 W_{1i}^{(2)} (W_{i1}^{(1)} x_1 + W_{i2}^{(1)} x_2)^2$, where $ W_{ij}$ is the $(i,j)$ entry of $W$. \
> Set another network with the same architecture as \
> $\tilde{f} = \sum_{i=1}^3 V_{1i}^{(2)} (V_{i1}^{(1)} x_1 + V_{i2}^{(1)} x_2)^2$ using weights $V^{(1)} \in \mathbb{R}^{3\times 2}, V^{(2)} \in \mathbb{R}^{1\times 3}$. \
> The condition for the two networks modeling identical functions is as follows: \
> $\sum_{i=1}^3 W_{1i}^{(2)} (W_{i1}^{(1)} x_1 + W_{i2}^{(1)} x_2)^2 = \sum_{i=1}^3 V_{1i}^{(2)} (V_{i1}^{(1)} x_1 + V_{i2}^{(1)} x_2)^2$ for all $x \in \mathbb{R}^2$. \
> If we rearrange the above for each coefficient of $ x_1^2, x_1 x_2$, and $x_2^2$,
> we obtain \
> (i) $\sum_{i=1}^3 W_{1i}^{(2)} (W_{i1}^{(1)})^2 = \sum_{i=1}^3 V_{1i}^{(2)} (V_{i1}^{(1)})^2,$ \
> (ii) $\sum_{i=1}^3 W_{1i}^{(2)} W_{i1}^{(1)} W_{i2}^{(1)} = \sum_{i=1}^3 V_{1i}^{(2)} V_{i1}^{(1)} V_{i2}^{(1)},$ and \
> (iii) $\sum_{i=1}^3 W_{1i}^{(2)} (W_{i2}^{(1)})^2 = \sum_{i=1}^3 V_{1i}^{(2)} (V_{i2}^{(1)})^2$. \
> For the set of $ W^{(1)}, W^{(2)}, V^{(1)}, V^{(2)}$ that satisfy above equations, the two network functions become identical. \
> Consider finding $V^{(2)}$ for given $ W^{(1)}, W^{(2)}$, and $V^{(1)} = \phi_1(W^{(1)})$ for a bijective nonlinear function $\phi: \mathbb{R}^{3\times 2} \rightarrow \mathbb{R}^{3\times 2}$. Since there are three equations (i,ii,iii) and three variables of $(V_{11}^{(2)}, V_{12}^{(2)}, V_{13}^{(2)})$ in $V^{(2)}$, we can find the solution of $V^{(2)}$ and represent it as $V^{(2)} = \phi_2(W^{(1)}, W^{(2)})$. \
> We can then figure out that two different weight configuration with a nonlinear relationship between each other as  $V^{(1)} = \phi_1(W^{(1)})$ and $V^{(2)} = \phi_2(W^{(1)}, W^{(2)})$ can model identical network functions. \
> When the input or output dimension increases, if the hidden variable dimension increases appropriately, we can find mappings between weights that can model identical neural networks similarly.

---

### Official Review · Reviewer_Mksd · 2022-07-13

**Rating:** 7
**Confidence:** 4
**Soundness:** 3 good
**Presentation:** 3 good
**Contribution:** 3 good

**Summary:**

This paper presents a sharpness measure based on the Fisher information matrix of the probability distribution parameterized by a neural network. This measure takes into account reparameterizations other than just the rescaling of the weight matrices, and extends the invariance to any continuous transformation of the neural network parameters. The authors provide comparisons between their sharpness measures and other scale-invariant measures. They also run experiments showing that adding their measure as a regularizer can improve generalization.

**Questions:**

The entire development in the paper seems to be tied to the cross entropy loss. While that is standard for classification, the square loss is also widely used for classification. Is the IGS measure easily adaptable to that case?


**Limitations:**

Yes

**Strengths And Weaknesses:**

The paper is well written and the sharpness measure is well motivated. The experiments are rigorous and the visualization of decision boundaries is informative.

Weaknesses:
The comparison against other scale-invariant sharpness measures is illustrative but a little unfair since those measures were only designed to be invariant to scale (especially the one based on the quotient manifold).

The sections on theoretical connections to generalization only show how the IGS measure can be plugged into other criteria that are correlated with generalization. This paper does not answer the larger question of how sharpness can explain generalization - and whether this is a phenomenon unique to deep learning, or whether it explains generalization in other statistical learning problems.

---

> ### Author Response · Authors · 2022-08-01
> **Reply to reviewer's comments**
>
> ### About weaknesses
> * The reviewer's concern regarding comparison with other scale-invariant sharpness measures is understandable. However, it was inevitable to perform such experiments to confirm the reparametrization invariance, a unique property of our sharpness measure that most existing sharpness measures did not have.
> * We would like to clarify that TIC, which our measure can be approximated to, is indeed theoretically linked to the generalization gap beyond its empirical correlation with generalization. Under asymptotic assumptions, TIC theoretically estimates the bias $b(\hat{\theta})$ of the maximum log-likelihood value, i.e., TIC = $\lim\limits_{N \to \infty} b(\hat{\theta})$ with $N$ as the number of training data. The bias is defined as follows:\
> $ b(\hat{\theta}) = E_{\mathcal{D}}\left[ \frac{1}{\left|\mathcal{D}\right|}\sum\limits_{(x_i, y_i) \in \mathcal{D}}\log p(x_i, y_i; \hat{\theta}(\mathcal{D})) – E_{(x,y)}\left[ \log p(x, y; \hat{\theta}(\mathcal{D}))\right]\right]$, where $\mathcal{D}$ = $\\{ (x_1, y_1), \ldots, (x_N, y_N) \\}$ is the training data with $(x_i, y_i) \sim p(x,y)$ and $\hat{\theta}(\mathcal{D})$ denotes the maximum likelihood estimate obtained from using $\mathcal{D}$ [1'].
> Since our loss is the negative log-likelihood, the first and second term inside $E_{\mathcal{D}}[\cdot]$ corresponds to the (negative) training and test losses, respectively. Consequently, it can be said that TIC estimates the expected generalization gap (i.e., test loss – training loss) of the parametric model. \
> Even so, as the reviewer noted, it would be difficult to say that our study provides a complete answer to the larger question of how sharpness can explain generalization. However, in the process of finding answers to these conundrums, we expect that our measures and our concept of applying information geometry would give useful intuition and motivate subsequent studies. \
> As for whether sharpness can be applied to learning problems other than deep learning, at least for our measure, it has a strong connection with the above-mentioned TIC. Since TIC can be defined for a general parametric model, there would be ample room for our measure to be applied to other parametric models. However, further research is needed to verify whether diverse sharpness concepts can be applied to other statistical learning problems.
>
> ### Answers to questions
> * Our measure can also be applied to the case of squared loss. Using the log-likelihood model corresponding to the squared loss and the corresponding FIM, the definition of our measure, reparametrization- and scale-invariance properties, and the connections with generalization in terms of TIC and margin can all be seen as well. \
> More specifically, the invariance properties are satisfied by definition. The link to the TIC also comes from the fact that the Fisher information matrix and the Hessian are approximately equal when the neural network is sufficiently trained for the squared loss [2']. In terms of our definition of margin, if we let $l_i = \frac{1}{2}(z_i – f(x_i; \theta))^2$ (with $z_i \in \{0, 1\}$) and set the condition for data lying on the decision boundary as $f(x; \theta) = \frac{1}{2}$, we can show the connection of the measure to the margin without much difficulty according to a discussion similar to Section 3.3.2.
>
> [1'] Dixon, M., Ward, T.: Takeuchi’s information criteria as a form of regularization. arXiv preprint arXiv:1803.04947 (2018)
>
> [2'] Martens, J.: New insights and perspectives on the natural gradient method. Journal of Machine Learning Research 21, 1–76 (2020)

---

### Meta-Review · Area_Chair_Zzwg · 2022-08-27

**Recommendation:** Accept
**Confidence:** Certain

**Metareview:**

The expert reviewers appreciated the contributions and ideas in this paper and also liked their rebuttals.
It is also good to see new ideas being introduced to the important topic of flatness measures for deep learning.
Also, the presentation also enhances the paper's reputation.

In contrast, it is worth noting that their claims are somewhat excessive. The reviewers have pointed out specific modifications, hence I recommend to accept this paper on the condition that the points are modified.

**Award:**

No

---

### Decision · Program_Chairs · 2022-09-14

Accept